# ANALYZING AND EVALUATING UNBIASED LANGUAGE MODEL WATERMARK

**Yihan Wu**[*], **Xuehao Cui**[*], **Ruibo Chen**, **Heng Huang**[†]
Department of Computer Science
University of Maryland, College Park
{`ywu42, cedrus, rbchen, heng`}@umd.edu

## ABSTRACT

Verifying the authenticity of AI-generated text has become increasingly important with the rapid advancement of large language models, and unbiased watermarking has emerged as a promising approach due to its ability to preserve output distribution without degrading quality. However, recent work reveals that unbiased watermarks can accumulate distributional bias over multiple generations and that existing robustness evaluations are inconsistent across studies. To address these issues, we introduce UWBENCH, the first open-source benchmark dedicated to the principled evaluation of unbiased watermarking methods. Our framework combines theoretical and empirical contributions: we propose a statistical metric to quantify multi-batch distribution drift, prove an impossibility result showing that no unbiased watermark can perfectly preserve the distribution under infinite queries, and develop a formal analysis of robustness against token-level modification attacks. Complementing this theory, we establish a three-axis evaluation protocol—unbiasedness, detectability, and robustness—and show that token modification attacks provide more stable robustness assessments than paraphrasing-based methods. Together, UWBENCH offers the community a standardized and reproducible platform for advancing the design and evaluation of unbiased watermarking algorithms. Codes are available at `https://github.com/cavosamir/UWBench.git`.

## 1 INTRODUCTION

As the capabilities of large language models have grown significantly in recent years, verifying the authenticity and origin of AI-generated content has become increasingly critical. Watermarking language models (Aaronson, 2022; Kirchenbauer et al., 2023a; Christ et al., 2023; Kuditipudi et al., 2023; Hu et al., 2023; Wu et al., 2023; Chen et al., 2024a;b; 2025; Mao et al., 2024; Dathathri et al., 2024) has emerged as a promising solution for distinguishing machine-generated text from human-authored content. These methods embed covert statistical signals into the generation process using specific keys, allowing downstream detection via statistical hypothesis testing to verify authorship without degrading fluency.

A particularly important class of these methods is unbiased watermarking, which aims to preserve the original distribution of the language model's outputs. Such methods are crucial for practical deployment since they do not introduce detectable distortions or degrade generation quality (Aaronson, 2022; Christ et al., 2023; Kuditipudi et al., 2023; Hu et al., 2023; Wu et al., 2023; Chen et al., 2025; Mao et al., 2024; Dathathri et al., 2024). However, recent studies have revealed important limitations. While unbiased watermarks may preserve the output distribution in expectation, their statistical properties can drift over multiple generations, leading to distribution bias that violates the original unbiasedness guarantees (Christ et al., 2023; Kuditipudi et al., 2023; Hu et al., 2023). Moreover, robustness evaluations in prior work are fragmented: different methods are tested against different adversaries using inconsistent protocols, leaving a gap in standardized, comparable assessment.

---

[*]Equal contribution

[†]This work was partially supported by NSF IIS 2347592, 2348169, DBI 2405416, CCF 2348306, CNS 2347617, RISE 2536663.

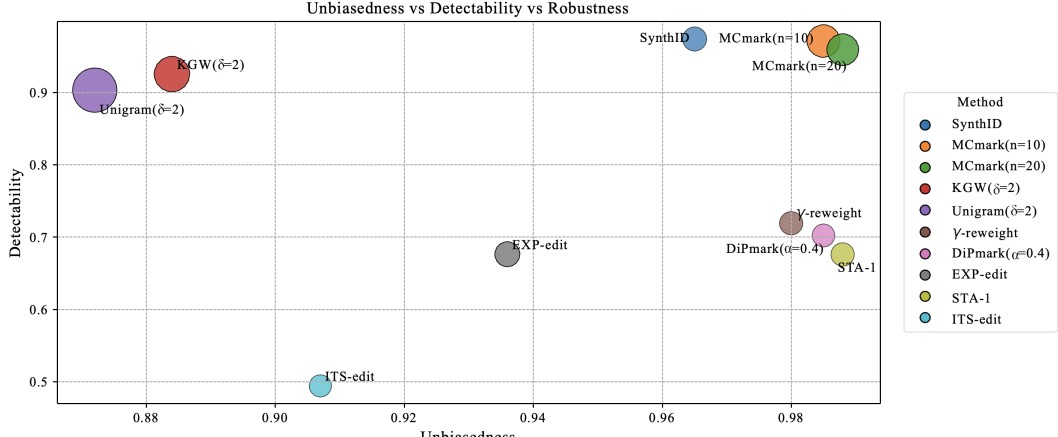

Figure 1: Overall benchmarking results of unbiasedness (x-axis), detectability (y-axis), and robustness (encoded with marker size) on different language model watermarking methods. Points further to the right and higher indicate better unbiasedness and detectability; larger markers indicate greater robustness.

To address these challenges, we introduce UWBENCH, the first open-source benchmark specifically designed for the analysis and evaluation of unbiased watermarking algorithms. Our framework offers both theoretical foundations and practical tools to facilitate principled comparisons. On the theoretical front, we propose a statistical metric that quantifies distributional shift across batches of generated texts, enabling evaluation of long-term bias. We further prove a general impossibility result: no unbiased watermark can perfectly preserve the model's output distribution under an infinite query budget. Finally, we develop a formal framework for analyzing the robustness of unbiased watermarking algorithms against token-level modification attacks, showing that such attacks can be resisted under certain structural assumptions.

In addition to the theoretical contributions, we provide a comprehensive empirical toolkit for benchmarking existing and future unbiased watermarking algorithms. We establish a three-axis evaluation protocol—unbiasedness, detectability, and robustness—that provides a holistic view of watermark performance. Notably, we revisit common adversarial attacks and demonstrate that paraphrasing-based evaluations suffer from high variance and inconsistent results, potentially leading to misleading conclusions. In contrast, token modification attacks yield more stable and reliable robustness assessments, making them a preferred choice for empirical benchmarking.

Our main contributions are summarized as follows:

- We introduce UWBENCH, an open-source benchmark designed specifically for evaluating unbiased watermarking methods in language models, with support for systematic and reproducible comparisons.

- We propose a multi-batch distribution bias metric and prove a fundamental limitation: no unbiased watermark can preserve the model's output distribution under unlimited queries. We also develop a theoretical framework for analyzing robustness against token-level attacks.

- We establish a three-axis evaluation protocol—unbiasedness, detectability, and robustness—and show that token modification attacks offer more stable and reliable robustness assessments than paraphrasing-based attacks.

## 2 RELATED WORK

**Statistical watermarks.** Kirchenbauer et al. (2023a) enhanced the statistical watermark framework originally introduced by Aaronson (2022), demonstrating the effectiveness of statistical watermarking through extensive experiments on large language models. They splited the LM tokens into red and green list, then promoted the use of green tokens by adding a fixed parameter $\delta$ to their logits. Zhao et al. (2023) proposed the unigram watermark, which enhances the robustness of the statistical watermark by using one-gram hashing to produce watermark keys. Liu et al. (2023b) also improved

the robustness of statistical watermarking by leveraging the semantics of generated content as watermark keys. Additionally, Liu et al. (2023a) proposed an unforgeable watermark scheme that employs neural networks to modify token distributions instead of using traditional watermark keys. However, these approaches may lead to significant changes in the distribution of generated text, potentially compromising content quality.

**Unbiased watermarks.** To maintain the original output distribution in watermarked content, several researchers have investigated novel approaches for token distribution modification. Aaronson (2022) pioneered an unbiased watermarking method using Gumbel-max to adjust token distribution and employing prefix n-grams as watermark keys. Christ et al. (2023) used inverse sampling for modifying the token distributions of watermarked content on a binary language model with watermark keys based on token positioning. ITS-edit and EXP-edit Kuditipudi et al. (2023) utilized inverse-sampling and Gumbel-max respectively for modifying the token distributions of watermarked content, with a predetermined watermark key list. Hu et al. (2023) combined inverse-sampling and $\gamma$-reweight strategies for watermarking, though their detection method is not model-agnostic. DiPmark Wu et al. (2023) enhanced the $\gamma$-reweight technique and introduced a model-agnostic detector. STA-1 Mao et al. (2024) optimized the quality of the watermarked text under the low-entropy scenarios. Dathathri et al. (2024) proposed SynthID, which enables distortion-freeness of LM watermarking with multiple generations. Chen et al. (2025) introduced MCMark, which significantly improved the detectability of the unbiased watermark. $MC^2Mark$ (Cui et al., 2026) further increased the watermark encoding capacity to enable multi-bit watermarking.

**LM watermarking benchmarks.** WaterBench (Tu et al., 2023) provides a comprehensive benchmark for LLM watermarking methods. It standardizes watermarking strength by tuning each method's hyperparameters to a common level, and then jointly evaluates both generation quality and detection performance. MarkMyWords (Piet et al., 2025) evaluates LLM watermarks along three dimensions: generation quality, detection efficiency (measured by the number of tokens required), and robustness. MarkLLM (Pan et al., 2024) introduces an open-source toolkit that offers a unified, extensible framework for implementing LLM watermarking algorithms, along with user-friendly interfaces to facilitate broader adoption. However, most of the watermarking methods covered in these benchmarks are biased. They lack a thorough analysis of unbiased watermarking techniques, and do not include evaluation metrics specifically designed for them. As such, we argue that a dedicated benchmark for unbiased watermarking is both necessary and timely.

# 3 PRELIMINARY

## 3.1 MOTIVATION

Statistical watermarking has emerged as a general-purpose solution for verifying the authenticity of AI-generated content. Unlike task-specific benchmarks, statistical watermarking can be applied to *any* language model and across *any* downstream task without the need for collecting task-dependent datasets. Thus, for evaluating unbiased watermarking methods, building a new dataset is unnecessary and does not address the core challenges. Instead, the true value of a benchmark lies in providing principled and reliable *metrics* for assessing watermark performance.

Current unbiased watermarking methods are typically evaluated along three axes: unbiasedness, detectability, and robustness. While detectability metrics are relatively well-established, existing approaches for measuring unbiasedness and robustness are inadequate. In particular, unbiasedness has so far been evaluated under a single-prompt setting, which overlooks important failure cases. We theoretically prove a fundamental impossibility: no watermarking scheme can remain unbiased when the same prompt is repeatedly queried. Motivated by this result, we propose a new metric that quantifies distributional bias under repeated queries, offering a more faithful measure of unbiasedness.

Robustness evaluation presents another challenge. Most existing work relies on paraphrasing-based adversarial attacks. However, these methods suffer from high variance and inconsistent results (See Figure 2), leading to unreliable conclusions. To overcome this limitation, we combine the paraphrasing-based adversarial attacks with the random token modification attacks that provides stable and reproducible assessments robustness.

In summary, UW Bench shifts the focus of watermarking evaluation away from task-specific datasets and toward theoretically grounded, reproducible, and holistic performance metrics that better capture the limitations and strengths of unbiased watermarking algorithms.

## 3.2 WATERMARKING SETTING

**Problem Definition.** A language model (LM) provider aims to watermark generated text so that any user can later verify its origin, without access to the LM or the original prompt. A watermarking framework consists of two components: a *watermark generator* and a *watermark detector*. The generator embeds hidden statistical signals into the text, while the detector recovers these signals using hypothesis testing.

**Watermark Generator.** Let $P_M(\cdot \mid \boldsymbol{x}_{1:n})$ denote the LM's distribution for predicting the $n$-th token given prefix $\boldsymbol{x}_{1:n}$. A watermark key $k \in K$ and a reweight strategy $F$ are used to construct the watermarked distribution $P_W(\cdot \mid \boldsymbol{x}_{1:n}, k) = F\big(P_M(\cdot \mid \boldsymbol{x}_{1:n}), k\big)$. The next token $x_n$ is then sampled from $P_W$ instead of $P_M$. The watermark key typically includes a *secret key* sk and a *context key* (e.g., $n$-gram index (Aaronson, 2022) or token position (Christ et al., 2023)). This process injects a subtle statistical signal into the generated text.

The reweight strategy is the core of watermark generation. A strategy is called *distortion-free* if the resulting $P_W$ preserves the original distribution $P_M$. To date, three main families of distortion-free strategies have been proposed: (i) inverse-sampling (Christ et al., 2023; Kuditipudi et al., 2023; Hu et al., 2023), (ii) Gumbel-reparametrization (Aaronson, 2022; Kuditipudi et al., 2023), and (iii) permute-reweight (Hu et al., 2023).

**Definition (Unbiased Watermark).** A watermarking scheme is *unbiased* if, for any context $\boldsymbol{x}_{1:n}$, the expected distribution of the next token under watermarking matches the original LM distribution:

$$\mathbb{E}_{k \sim \mu}[\,P_W(\cdot \mid \boldsymbol{x}_{1:n}, k)\,] = P_M(\cdot \mid \boldsymbol{x}_{1:n}),$$

Where $\mu$ is the watermark key distribution. In other words, averaging over random watermark keys does not introduce any systematic distortion into the model's output distribution.

**Watermark Detector.** The detector only has access to the watermark key $k$ and the reweight strategy $F$. Detection is posed as a hypothesis test: $H_0$ : Text is unwatermarked, $H_1$ : Text is watermarked. To test this, a score function $s : V \times K \times \mathcal{F} \to \mathbb{R}$ is applied token by token. For a sequence $\boldsymbol{x}_{1:n}$, the test statistic is $S(\boldsymbol{x}_{1:n}) = \sum_{i=1}^{n} s(x_i, k, F)$. If $S(\boldsymbol{x}_{1:n})$ significantly deviates from its expected value under $H_0$, the null hypothesis is rejected and the text is declared watermarked.

## 4 UWBENCH

### 4.1 UNBIASEDNESS UNDER REPEATED PROMPTS

**Unbiasedness (one-shot).** Let $P_M(\cdot \mid \boldsymbol{x})$ be the LM distribution for prompt $\boldsymbol{x}$ and let $P_W(\cdot \mid \boldsymbol{x}, k)$ be the distribution induced by a watermark with key $k \sim \mu$. We say the watermark is *unbiased* in the one-shot sense if

$$\mathbb{E}_{k \sim \mu}[P_W(\cdot \mid \boldsymbol{x}, k)] \;=\; P_M(\cdot \mid \boldsymbol{x}) \quad \text{for all prompts } \boldsymbol{x}. \tag{1}$$

**Impossibility under repeated prompts.** We next state our main impossibility for repeated queries of the *same* prompt under a fixed key (proof deferred to the appendix).

**Theorem 4.1** (Unbiasedness breaks under repeated prompts). *No watermarking scheme can simultaneously satisfy: i) preservation of the original LM distribution under repeated queries of the same prompt with a fixed key $k$, and ii) detectability. Equivalently, any detectable scheme that is unbiased in the one-shot sense equation 4 fails to preserve $P_M$ when the same prompt is queried repeatedly under a fixed key.*

**Single-prompt multi-generation (SPMG) unbiasedness metric.** Theorem 4.1 motivates measuring distributional deviation when a single prompt is queried multiple times with a *fixed* key. Let $p_1, \ldots, p_n$ be $n$ prompts. For each $p_i$, draw $m$ independent generations from a model $P$ (fixing decoding settings; for watermarks, the key is held fixed across the $m$ draws). Let $\mathrm{Met}(\cdot)$ be any

bounded per-generation performance surrogate (e.g., perplexity, average log-likelihood, reward score), with $|\mathrm{Met}(g)| \leq A$. Define the per-prompt SPMG mean: $\overline{\mathrm{Met}}_i(P) := \frac{1}{m} \sum_{j=1}^{m} \mathrm{Met}(g_j^{p_i}(P))$, and the *SPMG gap* between two models $P$ and $Q$: $\Delta \mathrm{Met}(P, Q) := \frac{1}{n} \sum_{i=1}^{n} \left| \overline{\mathrm{Met}}_i(P) - \overline{\mathrm{Met}}_i(Q) \right|$. Intuitively, $\Delta \mathrm{Met}(P_M, P_T)$ captures the multi-sample deviation of a test model $P_T$ from the original $P_M$ that emerges only when the same prompt is queried repeatedly.

**Variance-controlled detection statistic.** To factor out natural sampling noise, we compare the test model to an *i.i.d. clone* of the original model. Let $P_{M'}$ be an independent model with the same distribution as $P_M$. Define the detection statistic

$$\mathrm{DetWmk}(P_M, P_T) := \Delta \mathrm{Met}(P_M, P_T) - \Delta \mathrm{Met}(P_M, P_{M'}).$$

Large positive values indicate a repeated-prompt shift beyond the intrinsic variance of $P_M$.

**Theorem 4.2** (McDiarmid concentration for for SPMG detection). *Suppose $P_T$ is identically distributed with $P_M$ and $|\mathrm{Met}(g)| \leq A$ for all generations $g$. Then for any $t > 0$,*

$$\Pr\Big( \big| \mathrm{DetWmk}(P_M, P_T) - \mathbb{E}[\mathrm{DetWmk}(P_M, P_T)] \big| \geq t \Big) \leq 2 \exp\Big( -\frac{mn\,t^2}{12A^2} \Big). \tag{2}$$

*Equivalently, with probability at least $1 - \delta$,*

$$\big| \mathrm{DetWmk}(P_M, P_T) - \mathbb{E}[\mathrm{DetWmk}(P_M, P_T)] \big| \leq A \sqrt{\frac{12 \log(2/\delta)}{mn}}.$$

Inequality 2 yields an $\alpha$-level threshold $t_\alpha = A^2 \sqrt{\frac{12 \ln(1/\alpha)}{mn}}$ to control false positives when testing for repeated-prompt bias. Consequently, SPMG-based evaluation isolates the distributional drift that Theorem 4.1 predicts, while providing finite-sample guarantees for reliable detection.

## 4.2 ROBUSTNESS ANALYSIS OF UNBIASED WATERMARKS

**Adversary model.** During detection, only the text sequence is available to the verifier; hence an adversary can act solely by *modifying tokens*. We consider an edit-bounded adversary that applies up to $b$ token operations (substitution/insertion/deletion), producing an attacked sequence $\boldsymbol{x}'$. Let the detector use an additive test statistic $S(\boldsymbol{x}) = \sum_{t=1}^{T} s_t(\boldsymbol{x})$ with decision threshold $\tau$ (reject $H_0$ if $S(\boldsymbol{x}) \geq \tau$). All scores are assumed bounded: $s_t(\boldsymbol{x}) \in [0, B]$.

**Limitations of existing attack protocols.** Prior works evaluate robustness with random token edits (Kirchenbauer et al., 2023a;b), paraphrasing (Kirchenbauer et al., 2023b), and translation (He et al., 2024). These are imperfect for benchmarking: (i) random edits often severely degrade semantic quality; (ii) paraphrasing exhibits instability across prompts and seeds; (iii) translation is *too strong*: it changes essentially all tokens, so no unbiased watermark can survive, making methods indistinguishable. This motivates a principled, token-level robustness characterization with *certificates*.

**Token effect region.** Let $\mathcal{C}_t(\boldsymbol{x})$ denote the context used by the detector to score token $t$ (e.g., an $n$-gram prefix or a rolling, prefix-dependent key schedule). A modification at position $i$ can affect the scores for all tokens $t$ whose context uses $x_i$, i.e., $\{t : x_i \in \mathcal{C}_t(\boldsymbol{x})\}$. Define the *token effect region length* of position $i$ by $R_i(\boldsymbol{x}) := \big| \{t \geq i : x_i \in \mathcal{C}_t(\boldsymbol{x})\} \big|$. For detectors keyed by an $n$-gram prefix, $R_i(\boldsymbol{x}) \leq n + 1$ (only $t \in [i, i+n]$ are influenced). For position-key schedules that depend on the entire prefix (rolling hash), $R_i(\boldsymbol{x}) = T - i + 1$ (all suffix tokens can be influenced). Let $R_{\max} := \max_i R_i(\boldsymbol{x})$.

**Expected score decrease under one edit.** Write the detector as $S(\boldsymbol{x}) = \sum_{t=1}^{T} s_t(\boldsymbol{x})$ and let $\Delta_i$ denote the *expected* reduction in $S$ caused by editing token $i$, where the expectation is taken over the randomized alignment (e.g., color assignment or bit tests) induced when the context is destroyed in the affected region. Then $\mathbb{E}\big[ S(\boldsymbol{x}) - S(\boldsymbol{x}^{(i)}) \big] = \sum_{t : x_i \in \mathcal{C}_t(\boldsymbol{x})} \big( \mathbb{E}[s_t(\boldsymbol{x})] - \mathbb{E}[s_t(\boldsymbol{x}^{(i)})] \big)$. Instantiations for common unbiased watermark families:

**Green-count detectors** (e.g., $\gamma$-reweight, DiPmark, STA): $s_t \in \{0, 1\}$ indicates whether token $t$ falls in the *green* set. Let $P_G$ be the (empirical) fraction of green tokens under watermarking. Destroying alignment in the effect region makes green assignment effectively random, yielding an expected per-token drop of $(2P_G - 1)/2$. Hence for one edit with effect length $R$, $\mathbb{E}\big[S(\boldsymbol{x}) - S(\boldsymbol{x}^{(i)})\big] = \frac{(2P_G-1)}{2} R$.

**SynthID-style bit tests:** Each token contributes $m$ binary scores, $s_t = \sum_{\ell=1}^m s_{t,\ell}$ with $s_{t,\ell} \in \{0, 1\}$. Let $P_s := \mathbb{E}[s_t]$ under watermarking. Randomized alignment drives each bit to mean $1/2$, so the expected per-token drop is $(P_s - \frac{m}{2})$, yielding $\mathbb{E}\big[S(\boldsymbol{x}) - S(\boldsymbol{x}^{(i)})\big] = (P_s - \frac{m}{2}) R$.

**Certified robustness.** Because each single-token edit can affect at most $R_{\max}$ token scores and each token score changes by at most $B$, the test statistic is *Lipschitz* w.r.t. edit distance $S(\boldsymbol{x}) - S(\boldsymbol{x}') \leq b R_{\max} B$ for any $b$-edit attack. Hence we obtain an $\ell_0$ *certified radius*:

$$S(\boldsymbol{x}) - \tau > b R_{\max} B \implies S(\boldsymbol{x}') \geq \tau \quad \text{for all } \boldsymbol{x}' \text{ with } \leq b \text{ edits.} \qquad (3)$$

This bound holds without distributional assumptions (worst-case guarantee).

## 5 EXPERIMENTS

Our evaluation is organized along three axes. *(i) Unbiasedness:* we measure watermarking unbiasedness in one-shot settings (machine translation and text summarization tasks; BLEU/ROUGE/BERTScore) and quantify repeated-prompt distribution shift via the SPMG metrics $\Delta\text{Met}$ and the calibrated statistic $\text{DetWmk}$. *(ii) Detectability:* on open-ended generation (C4/MMW/Dolly CW/WaterBench) we report TPR at theoretically guaranteed FPR levels $(5\%, 1\%, 0.1\%)$ and AUC using matched watermarked/unwatermarked sets across Llama-3.2-3B-Instruct, Mistral-7B-Instruct-v0.3, and Phi-3.5-mini-instruct. *(iii) Robustness:* We use paraphrasing attack and random token modification under edit budgets. Detailed setups and hyperparameters are in Appendix C.

**Baselines.** We compare against representative *unbiased* watermarking algorithms: $\gamma$-reweight (Hu et al., 2023), DiPmark (Wu et al., 2023), MCmark (Chen et al., 2025), SynthID (Dathathri et al., 2024), ITS-Edit (Kuditipudi et al., 2023), EXP-Edit (Kuditipudi et al., 2023), and STA-1 (Mao et al., 2024). Besides, we add two popular biased watermark: KGW Kirchenbauer et al. (2023a) and Unigram Zhao et al. (2023) as additional baselines.

**Unbiasedness.** Following Hu et al. (2023); Wu et al. (2023), we compare task metrics between the original LM and its watermarked counterpart: Machine translation: BLEU and BERTScore on WMT16 RO-EN; Text summarization: ROUGE-1/2/L and BERTScore on CNN/DAILYMAIL. We evaluate with (a) 1000 prompts, one generation per prompt (Table 1); and (b) 10 prompts, 1000 generations per prompt (SPMG, Table 2). To measure repeated-prompt bias, we adopt the SMPG gap $\Delta\text{Met}(P, Q)$ and report the calibrated statistic $\text{DetWmk}(P_M, P_T) := \Delta\text{Met}(P_M, P_T) - \Delta\text{Met}(P_M, P_{M'})$ with bounded $\text{Met}$ (e.g. perplexity, or bounded quality scores).

**Detectability.** Open-ended generation on C4/MMW/DOLLY CW/WATERBENCH: 1000 prompts, one generation per prompt. For each method we build matched sets of watermarked and unwatermarked texts (same prompts, decoding settings). We compute: (i) TPR at target FPR $\{5\%, 1\%, 0.1\%\}$ using analytic thresholds from each detector's null; (ii) Median p-value generated by the detection algorithm; (iii) AUC on balanced datasets (same number of positive/negative sequences). Unless otherwise stated, we fix generation lengths around 500 tokens per dataset.

**Robustness.** We evaluate watermark robustness under two categories of paraphrasing-based attacks. DIPPER is a strong neural paraphraser that rewrites text while preserving semantic meaning, thereby introducing substantial variability into the generated outputs. In contrast, Random token replacement attacks directly perturb the text by substituting a fixed percentage of tokens (10%, 20%, or 30%) with randomly sampled alternatives. While random replacements offer a simple, noise-driven baseline for robustness testing, DIPPER provides a more realistic and challenging paraphrasing scenario that better reflects practical adversarial conditions.

Table 1: Unbiasedness evaluation a) (1000 prompts, 1 generations each). We evaluate the unbiasedness of watermarking methods on text summarization and machine translation tasks.

| Method | Text Summarization | | | Machine Translation | |
|---|---|---|---|---|---|
| | BERTScore | ROUGE-1 | Perplexity | BERTScore | BLEU |
| No watermark | 0.3077 | 0.3807 | 6.39 | 0.5432 | 20.1681 |
| Unigram($\delta$=0.5) | 0.3080 | 0.3773 | 6.54 | 0.5436 | 20.0175 |
| Unigram($\delta$=1.0) | 0.3053 | 0.3775 | 6.85 | 0.5388 | 20.1276 |
| Unigram($\delta$=1.5) | 0.2955 | 0.3656 | 7.51 | 0.5307 | 19.5000 |
| Unigram($\delta$=2.0) | 0.2848 | 0.3566 | 8.28 | 0.5191 | 18.4838 |
| KGW($\delta$=0.5) | 0.3012 | 0.3757 | 6.52 | 0.5472 | 20.6198 |
| KGW($\delta$=1.0) | 0.2977 | 0.3751 | 6.85 | 0.5348 | 19.9166 |
| KGW($\delta$=1.5) | 0.2876 | 0.3686 | 7.56 | 0.5326 | 19.2318 |
| KGW($\delta$=2.0) | 0.2769 | 0.3619 | 8.37 | 0.5218 | 17.9401 |
| DIP($\alpha$=0.3) | 0.3082 | 0.3793 | 6.41 | 0.5422 | 20.2514 |
| DIP($\alpha$=0.4) | 0.3081 | 0.3781 | 6.53 | 0.5446 | 20.4579 |
| $\gamma$-reweight | 0.3032 | 0.3749 | 6.49 | 0.5394 | 20.5546 |
| MCmark(n=10) | 0.3032 | 0.3755 | 6.39 | 0.5416 | 20.4171 |
| MCmark(n=20) | 0.3054 | 0.3780 | 6.41 | 0.5486 | 20.0984 |
| MCmark(n=50) | 0.3099 | 0.3810 | 6.45 | 0.5400 | 20.1503 |
| MCmark(n=100) | 0.3080 | 0.3800 | 6.46 | 0.5466 | 20.6732 |
| STA-1 | 0.3066 | 0.3793 | 6.25 | 0.5492 | 20.5561 |
| SynthID | 0.3049 | 0.3775 | 6.37 | 0.5445 | 20.4107 |
| EXP-Edit | 0.3114 | 0.3797 | 6.19 | 0.5458 | 20.4879 |
| ITS-Edit | 0.3032 | 0.3749 | 6.58 | 0.5091 | 17.9904 |

Table 2: Unbiasedness evaluation b) (10 prompts, 1000 generations each)We evaluate the unbiasedness of watermarking methods on text summarization and machine translation tasks with **SPMG** metric.

| Method | Text Summarization | | | Machine Translation | |
|---|---|---|---|---|---|
| | BERTScore | ROUGE-1 | Perplexity | BERTScore | BLEU |
| No watermark | 0.0026 | 0.0017 | 0.1828 | 0.0033 | 0.1199 |
| Unigram($\delta$=0.5) | 0.0037 | 0.0033 | 0.2197 | 0.0074 | 0.8311 |
| Unigram($\delta$=1.0) | 0.0076 | 0.0076 | 0.6133 | 0.0181 | 1.7137 |
| Unigram($\delta$=1.5) | 0.0146 | 0.0148 | 1.3664 | 0.0293 | 2.8560 |
| Unigram($\delta$=2.0) | 0.0255 | 0.0256 | 2.4671 | 0.0423 | 3.9869 |
| KGW($\delta$=0.5) | 0.0040 | 0.0023 | 0.1051 | 0.0041 | 0.5211 |
| KGW($\delta$=1.0) | 0.0093 | 0.0062 | 0.4095 | 0.0106 | 1.1383 |
| KGW($\delta$=1.5) | 0.0177 | 0.0121 | 0.9434 | 0.0178 | 1.5189 |
| KGW($\delta$=2.0) | 0.0297 | 0.0199 | 1.9382 | 0.0232 | 2.0037 |
| DIP($\alpha$=0.3) | 0.0050 | 0.0039 | 0.0484 | 0.0147 | 1.2311 |
| DIP($\alpha$=0.4) | 0.0067 | 0.0059 | 0.1772 | 0.0149 | 1.6594 |
| $\gamma$-reweight | 0.0071 | 0.0081 | 0.1570 | 0.0174 | 1.9001 |
| MCmark(n=10) | 0.0073 | 0.0033 | 0.2456 | 0.0171 | 1.5756 |
| MCmark(n=20) | 0.0066 | 0.0037 | 0.2914 | 0.0162 | 0.9958 |
| MCmark(n=50) | 0.0069 | 0.0076 | 0.2771 | 0.0153 | 0.7727 |
| MCmark(n=100) | 0.0080 | 0.0077 | 0.3068 | 0.0234 | 0.6486 |
| STA-1 | 0.0046 | 0.0035 | 0.1505 | 0.0107 | 0.8446 |
| SynthID | 0.0159 | 0.0227 | 0.8254 | 0.0377 | 2.5266 |
| EXP-Edit | 0.0422 | 0.0413 | 2.0032 | 0.0439 | 2.4104 |
| ITS-Edit | 0.0355 | 0.0533 | 1.4912 | 0.0679 | 5.0746 |

**Paraphrasing variance.** Using DIPPER paraphrasing, we generate $r$ paraphrases per input (multiple seeds and temperatures), forming matched sets for each method. We report the mean $\pm$ std of TPR@FPR and AUC across seeds and show per-prompt variance distributions. As shown in Figure 2,

Table 3: Averaged detection performance across all language models and datasets by method. We also include two biased watermarks: KGW and Unigram for reference.

| Method | TPR@FPR=5% | TPR@FPR=1% | TPR@FPR=0.1% | median p-value | AUROC |
|---|---|---|---|---|---|
| Unigram($\delta$=0.5) | 68.7% | 53.59% | 35.55% | 3.72e-02 | 0.803 |
| Unigram($\delta$=1.0) | 90.04% | 81.57% | 69.49% | 3.00e-03 | 0.919 |
| Unigram($\delta$=1.5) | 96.57% | 93.07% | 86.96% | 5.56e-05 | 0.960 |
| Unigram($\delta$=2.0) | 98.89% | 97.85% | 94.66% | 1.05e-06 | 0.981 |
| KGW($\delta$=0.5) | 61.11% | 43.9% | 26.86% | 7.05e-02 | 0.863 |
| KGW($\delta$=1.0) | 87.07% | 79.04% | 68.6% | 8.70e-03 | 0.962 |
| KGW($\delta$=1.5) | 95.67% | 91.45% | 86.32% | 4.46e-04 | 0.987 |
| KGW($\delta$=2.0) | 98.33% | 96.57% | 94.04% | 1.07e-05 | 0.995 |
| DIP($\alpha$=0.3) | 78.92% | 69.26% | 58.11% | 2.03e-02 | 0.943 |
| DIP($\alpha$=0.4) | 82.61% | 74.11% | 64.73% | 1.33e-02 | 0.956 |
| $\gamma$-reweight | 83.68% | 75.85% | 66.43% | 9.14e-03 | 0.960 |
| EXP-Edit | 77.44% | 72.42% | 67.14% | 5.01e-02 | 0.906 |
| ITS-Edit | 55.11% | 48.29% | 41.67% | 1.43e-01 | 0.804 |
| MCmark(n=10) | 98.51% | 97.09% | 94.57% | 4.08e-06 | 0.993 |
| MCmark(n=100) | 95.32% | 92.2% | 87.53% | 5.66e-04 | 0.987 |
| MCmark(n=20) | 97.82% | 95.51% | 92.05% | 4.75e-05 | 0.994 |
| MCmark(n=50) | 97.25% | 95.38% | 91.66% | 9.56e-05 | 0.991 |
| STA-1 | 84.55% | 73.79% | 59.4% | 1.43e-02 | 0.953 |
| SynthID | 99.03% | 97.29% | 94.66% | 6.22e-06 | 0.995 |

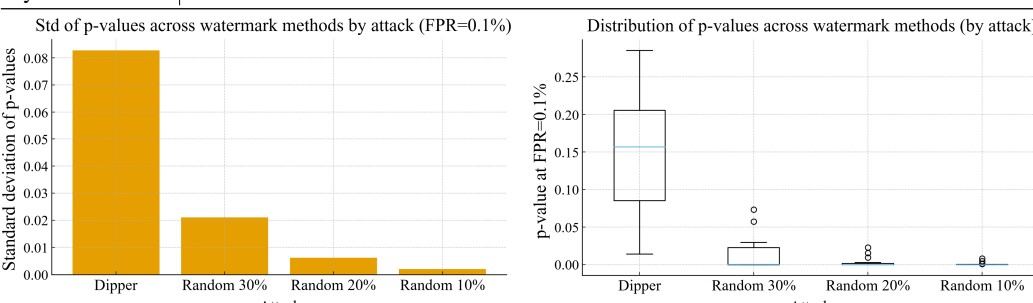

Figure 2: Variance of p-values across watermarking methods under different attack strategies (FPR=0.1%). Left: standard deviation of p-values by attack. Right: distribution of p-values (boxplots) across watermark methods.

DIPPER exhibits substantially higher variance in p-values compared to random token replacement attacks. The bar plot (left) shows that the standard deviation of p-values under DIPPER is roughly four times higher than under the strongest random attack (30% token replacement). The boxplot (right) further highlights this instability: DIPPER produces a wide spread of p-values, ranging from very low to relatively high values, while random replacements lead to consistently small p-values with much tighter distributions.

## 5.1 A THREE-AXIS EVALUATION OF UNBIASED WATERMARK

**Unbiasedness score.** For each method, we quantify unbiasedness as closeness to the unwatermarked baseline ("None") across metrics $m \in \{\text{TS-BERT}, \text{ROUGE-1}, \text{Perplexity}, \text{MT-BERT}, \text{BLEU}\}$. For Config 1, compute the relative deviation $r_m^{(1)} = |x_{m,\text{cfg1}}^{\text{method}} - x_{m,\text{cfg1}}^{\text{None}}|/x_{m,\text{cfg1}}^{\text{None}}$. For Config 2, treat reported values as deltas and remove the baseline noise floor via $r_m^{(2)} = \max\{0, |\Delta_{m,\text{cfg2}}^{\text{method}}| - |\Delta_{m,\text{cfg2}}^{\text{None}}|\}/x_{m,\text{cfg1}}^{\text{None}}$. Aggregate $D_1 = \frac{1}{M}\sum_m r_m^{(1)}$ and $D_2 = \frac{1}{M}\sum_m r_m^{(2)}$, then combine $D = \lambda D_1 + (1-\lambda)D_2$ (default $\lambda = 0.6$). Finally, map to $[0, 100]$ via the $100(1 - D)$ (default $\alpha = 1$); higher $U$ indicates greater unbiasedness (i.e., smaller average deviation from baseline and lower small-sample sensitivity).

**Detectability Score.** Using the averaged detection metrics per method (TPR at FPR $\in \{5\%, 1\%, 0.1\%\}$, median $p$-value, AUROC), first convert TPR percentages to decimals $tpr_5, tpr_1, tpr_{0.1} \in [0, 1]$ and form a low-FPR–weighted operating-point score $s_{\text{tpr}} = 0.2\,tpr_5 + 0.3\,tpr_1 + 0.5\,tpr_{0.1}$. Map median $p$-value to a bounded significance score via $s_p =$

Table 4: Robustness (TPR@1%FPR) of watermarking method across different attack types.

| Method | DIPPER | Random 30% | Random 20% | Random 10% |
|---|---|---|---|---|
| KGW($\delta$=0.5) | 0.42% | 4.26% | 7.55% | 8.19% |
| KGW($\delta$=1.0) | 0.21% | 14.90% | 24.48% | 38.75% |
| KGW($\delta$=1.5) | 1.46% | 37.60% | 56.77% | 75.10% |
| KGW($\delta$=2.0) | 1.35% | 62.11% | 78.32% | 91.68% |
| Unigram($\delta$=0.5) | 34.90% | 47.92% | 56.67% | 66.25% |
| Unigram($\delta$=1.0) | 40.52% | 64.38% | 80.52% | 85.00% |
| Unigram($\delta$=1.5) | 44.48% | 80.21% | 89.16% | 93.68% |
| Unigram($\delta$=2.0) | 58.85% | 95.63% | 98.02% | 99.17% |
| DIP($\alpha$=0.3) | 0.83% | 2.66% | 7.23% | 17.98% |
| DIP($\alpha$=0.4) | 0.42% | 3.37% | 7.37% | 21.16% |
| $\gamma$-reweight | 0.73% | 2.53% | 11.47% | 26.95% |
| STA($\gamma$=0.5) | 2.29% | 4.90% | 12.29% | 21.56% |
| SynthID | 3.02% | 7.71% | 14.58% | 26.25% |
| ITS-Edit | 1.15% | 2.40% | 3.96% | 6.04% |
| EXP-Edit | 0.94% | 15.21% | 21.46% | 26.98% |
| MCmark($n$=10) | 5.10% | 39.26% | 73.37% | 96.11% |
| MCmark($n$=20) | 3.85% | 33.85% | 61.56% | 86.25% |
| MCmark($n$=50) | 3.96% | 37.92% | 63.44% | 85.63% |
| MCmark($n$=100) | 3.02% | 30.42% | 50.52% | 71.15% |

$\min\{1, \ [-\log_{10}(\max(p, 10^{-22}))]/22\}$, which clips extremely small $p$ at $10^{-22}$ and yields $s_p \in [0, 1]$. Let $s_{\text{auc}} = \text{AUROC} \in [0, 1]$. The final detectability score is a convex combination

$$\text{Detect} = 100\big(w_{\text{tpr}}\, s_{\text{tpr}} + w_{\text{auc}}\, s_{\text{auc}} + w_p\, s_p\big),$$

with default weights $(w_{\text{tpr}}, w_{\text{auc}}, w_p) = (0.60, 0.25, 0.15)$. Higher values indicate stronger detectability, with emphasis on reliable detection at very low FPR while still rewarding overall separability (AUROC) and statistical significance (median $p$).

**Robustness score.** For each watermarking method $m$, let $t_{a,f}(m) \in [0, 1]$ denote the true positive rate (TPR, as a decimal) under attack $a \in \{\text{DIPPER}, \text{Random30}, \text{Random20}, \text{Random10}\}$ at false positive rate $f \in \{0.1\%, 1\%, 5\%\}$. We first compute a low-FPR–emphasized per-attack operating score $s_a(m) = 0.5\, t_{a,0.1\%}(m) + 0.3\, t_{a,1\%}(m) + 0.2\, t_{a,5\%}(m)$. These per-attack scores are then aggregated with reduced weight on DIPPER (reflecting the current study's priorities) using $(v_{\text{DIPPER}}, v_{\text{Random30}}, v_{\text{Random20}}, v_{\text{Random10}}) = (0.2, 0.4, \frac{4}{15} \approx 0.2667, \frac{2}{15} \approx 0.1333)$ to obtain a single robustness value $R(m) = \sum_a v_a\, s_a(m) \in [0, 1]$. The final non-smoothed robustness score reported in our tables is $\text{RobustnessScore}(m) = 100\, R(m) \in [0, 100]$; higher values indicate stronger robustness with greater emphasis on performance at very low FPR and under the more challenging random-replacement attacks.

Figure 2 and Table 7 jointly highlight the trade-offs between unbiasedness, detectability, and robustness across watermarking methods. From the scatter plot, we observe that methods such as MCmark ($n$=10/20) and SynthID occupy the top-right corner, demonstrating strong detectability and unbiasedness, though with limited robustness (small marker size). In contrast, Unigram ($\delta$=2) and KGW ($\delta$=2) achieve considerably higher robustness (large markers) but at the cost of lower unbiasedness and detectability. The tabulated scores further confirm this: Unigram ($\delta$=2) attains the highest robustness (0.855) despite relatively low detectability (0.903), whereas MCmark variants and SynthID provide balanced detectability (>0.945) and unbiasedness (>0.965) but modest robustness. Notably, DiPmark and STA-1 maintain excellent unbiasedness (>0.98) but their detectability lags behind (<0.72), highlighting their limitations under strict detection thresholds. Overall, these results underscore the central tension in watermark design: methods that optimize detectability and unbiasedness often sacrifice robustness, whereas highly robust methods compromise on unbiased generation quality or reliable detectability.

## 6 CONCLUSION

We introduced UWBENCH, an open-source benchmark for the principled evaluation of unbiased watermarking in language models. Our theory establishes a fundamental limitation: any detectable scheme that is unbiased in the one-shot sense cannot preserve the original distribution under repeated queries of the same prompt, motivating our single-prompt multiple-generation (SPMG) metric and a calibrated detection statistic for unbiasedness assessment. Experiments across diverse models and datasets demonstrate standardized, reproducible comparisons along three axes clarifying practical trade-offs and failure modes.

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

# A  LLM USAGE

We ONLY used ChatGPT-4o and ChatGPT-5 to refine the content.

# B  MISSING PROOFS

## B.1  PROOF OF THEOREM 4.1

**Setup.**  Fix a prompt $\boldsymbol{x}$. Let $P_M(\cdot \mid \boldsymbol{x})$ denote the LM's distribution over full generations (sequences or token paths). A watermarking scheme consists of a *reweight strategy* $F$ and a watermark key $k \in K$, producing a watermarked distribution

$$P_W(\cdot \mid \boldsymbol{x}, k) = F\big(P_M(\cdot \mid \boldsymbol{x}), k\big).$$

We say the scheme is *unbiased* (distribution-preserving in expectation over keys) if

$$\mathbb{E}_{k \sim \mu}[P_W(\cdot \mid \boldsymbol{x}, k)] = P_M(\cdot \mid \boldsymbol{x}), \tag{4}$$

where $\mu$ is the key distribution. Detectability means there exists a statistical test that, for some keys $k$, can distinguish samples from $P_W(\cdot \mid \boldsymbol{x}, k)$ versus $P_M(\cdot \mid \boldsymbol{x})$ with nontrivial power.

**Repeated-prompt model.**  Consider $m$ independent generations of the *same* prompt $\boldsymbol{x}$ under a *fixed* key $k$:

$$P_W^{(m)}(\cdot \mid \boldsymbol{x}, k) := \big(P_W(\cdot \mid \boldsymbol{x}, k)\big)^{\otimes m}, \qquad P_M^{(m)}(\cdot \mid \boldsymbol{x}) := \big(P_M(\cdot \mid \boldsymbol{x})\big)^{\otimes m}.$$

**Lemma B.1** (Detectability $\Rightarrow$ key-level deviation).  *If a watermarking scheme is detectable, then there exists a measurable set $A \subseteq K$ with $\mu(A) > 0$ such that $P_W(\cdot \mid \boldsymbol{x}, k) \neq P_M(\cdot \mid \boldsymbol{x})$ for all $k \in A$.*

*Proof.*  If $P_W(\cdot \mid \boldsymbol{x}, k) = P_M(\cdot \mid \boldsymbol{x})$ for $\mu$-almost every $k$, then for any sample size $m$ the product measures also coincide, $P_W^{(m)}(\cdot \mid \boldsymbol{x}, k) = P_M^{(m)}(\cdot \mid \boldsymbol{x})$, rendering any detector powerless (no test can outperform random guessing). Thus detectability implies a positive-measure subset of keys for which the two distributions differ. $\square$

**Lemma B.2** (Separation amplifies under products).  *Let $P \neq Q$ be two distributions on a common measurable space. Denote their Bhattacharyya coefficient by $\mathrm{BC}(P, Q) = \int \sqrt{dP\, dQ} \in (0, 1)$. Then for product measures,*

$$\mathrm{BC}\big(P^{\otimes m}, Q^{\otimes m}\big) = \big(\mathrm{BC}(P, Q)\big)^m \xrightarrow[m \to \infty]{} 0,$$

*and consequently the total variation distance satisfies*

$$\mathrm{TV}\big(P^{\otimes m}, Q^{\otimes m}\big) \geq 1 - \big(\mathrm{BC}(P, Q)\big)^m \xrightarrow[m \to \infty]{} 1.$$

*Proof.*  Bhattacharyya coefficients multiply under independent products. Using the inequality $1 - \mathrm{TV}(P, Q) \leq \mathrm{BC}(P, Q)$ yields the stated lower bound on TV; since $\mathrm{BC}(P, Q) < 1$ when $P \neq Q$, the bound tends to 1 as $m \to \infty$. $\square$

By Lemma B.1, detectability implies the existence of a positive-measure set $A$ of keys with $P_W(\cdot \mid \boldsymbol{x}, k) \neq P_M(\cdot \mid \boldsymbol{x})$. Fix any such $k \in A$ and apply Lemma B.2 with $P = P_W(\cdot \mid \boldsymbol{x}, k)$ and $Q = P_M(\cdot \mid \boldsymbol{x})$. Then

$$\mathrm{TV}\Big(P_W^{(m)}(\cdot \mid \boldsymbol{x}, k),\, P_M^{(m)}(\cdot \mid \boldsymbol{x})\Big) \xrightarrow[m \to \infty]{} 1,$$

so the product distributions diverge and become perfectly distinguishable as $m$ grows. Therefore, under repeated queries with a fixed key, the watermarked joint law cannot equal the LM's joint law; i.e., the scheme cannot preserve the original distribution under repeated prompts. This contradicts simultaneous satisfaction of (1)–(2) with detectability.

## B.2 PROOF OF THEOREM 4.2

*Proof.* Let $f$ map all $3nm$ sampled generations to $\text{DetWmk} = \Delta(P_M, P_T) - \Delta(P_M, P_{M'})$. Changing a single generation for prompt $i$ alters the corresponding per-prompt mean by at most $2A/m$, and since $x \mapsto |x - a|$ is 1-Lipschitz, the induced change on a $\Delta$ term is at most $(2A)/(mn)$.

*Bounded differences:*

- One $P_M$ sample affects both $\Delta(P_M, P_T)$ and $\Delta(P_M, P_{M'})$ by at most $(2A)/(mn)$ each, hence $|\Delta f| \leq (4A)/(mn)$.

- One $P_T$ sample affects only $\Delta(P_M, P_T)$: $|\Delta f| \leq (2A)/(mn)$.

- One $P_{M'}$ sample affects only $\Delta(P_M, P_{M'})$: $|\Delta f| \leq (2A)/(mn)$.

Summing squared Lipschitz constants over all variables gives

$$\sum_k c_k^2 = nm\left(\frac{4A}{mn}\right)^2 + nm\left(\frac{2A}{mn}\right)^2 + nm\left(\frac{2A}{mn}\right)^2 = \frac{24A^2}{mn}.$$

By McDiarmid's inequality,

$$\Pr\left(f - \mathbb{E}f \geq t\right) \leq \exp\left(-\frac{2t^2}{\sum_k c_k^2}\right) = \exp\left(-\frac{mn\,t^2}{12A^2}\right),$$

and the two-sided version follows by symmetry.

$\square$

## C   DETAILED EXPERIMENT SETUP

### C.1   EXPERIMENT SETUP

**Models & Datasets.**   We evaluate on LLAMA-3.2-3B-INSTRUCT (Dubey et al., 2024), MISTRAL-7B-INSTRUCT-V0.3 (Jiang et al., 2023), and PHI-3.5-MINI-INSTRUCT (Abdin et al., 2024) for open-ended text generation following prior work (Kirchenbauer et al., 2023a; Hu et al., 2023). Our primary corpus is a standard subset of C4 (Raffel et al., 2020); we additionally report on three MMW datasets (Piet et al., 2023), DOLLY CW (Conover et al., 2023), and two WATERBENCH tasks (Tu et al., 2023). For one-shot unbiasedness validation, we follow Hu et al. (2023); Wu et al. (2023) using MBART (Liu et al., 2020) on WMT16 RO-EN (Bojar et al., 2016) (machine translation) and BART (Lewis, 2019) on CNN/DAILYMAIL (See et al., 2017) (summarization).

**Watermarking setup.**   Unless noted, watermark keys combine a *secret key* with a *prefix 2-gram* context key. Hyperparameters follow the original papers: KGW $\delta \in 0.5, 1.0, 1.5, 2.0$, Unigram $\delta \in 0.5, 1.0, 1.5, 2.0$, DiPmark $\alpha \in \{0.3, 0.4\}$; SynthID tournament layers $m = 20$; MCmark list length $l \in 10, 20, 50, 100$; $\gamma$-reweight as in Hu et al. (2023). We report **TPR@FPR** at theoretically guaranteed FPR levels $\{5\%, 1\%, 0.1\%\}$ and **Median $p$-value**. Unless specified, decoding settings and prompt sets are identical across methods.

**Evaluation Metrics for Text Quality.** We employ the following metrics to assess the quality of generated text:

- **ROUGE.** For summarization tasks, we use the ROUGE metric (Lin, 2004), which measures n-gram overlap between generated summaries and reference texts, thereby capturing how well the output conveys the essential content.

- **BLEU.** For machine translation, we adopt the BLEU score (Papineni et al., 2002), which evaluates lexical similarity between system-generated translations and human references.

- **BERTScore.** BERTScore (Zhang et al., 2019) computes sentence similarity by aggregating cosine similarities between contextualized token embeddings. We report BERTScore-F1, -Precision, and -Recall for both summarization and translation tasks.

- **Perplexity.** Perplexity, a standard measure from information theory, quantifies how well a probabilistic model predicts observed text. Lower values indicate more accurate predictive performance. We use perplexity to evaluate both summarization and open-ended text generation.

# D ADDITIONAL RESULTS

Table 5: Robustness (TPR@0.1%FPR) of watermarking method across different attack types.

| Method | DIPPER | Random 30% | Random 20% | Random 10% |
|---|---|---|---|---|
| KGW($\delta$=0.5) | 0.00% | 2.23% | 2.45% | 4.26% |
| KGW($\delta$=1.0) | 0.00% | 4.69% | 9.69% | 18.75% |
| KGW($\delta$=1.5) | 0.31% | 17.29% | 35.00% | 50.94% |
| KGW($\delta$=2.0) | 0.00% | 40.21% | 62.21% | 80.53% |
| Unigram($\delta$=0.5) | 17.71% | 25.21% | 36.04% | 43.65% |
| Unigram($\delta$=1.0) | 21.35% | 41.88% | 55.52% | 69.06% |
| Unigram($\delta$=1.5) | 23.23% | 61.58% | 75.26% | 85.16% |
| Unigram($\delta$=2.0) | 36.46% | 86.88% | 93.23% | 95.83% |
| DIP($\alpha$=0.3) | 0.10% | 0.96% | 1.70% | 6.28% |
| DIP($\alpha$=0.4) | 0.10% | 0.74% | 3.79% | 8.21% |
| $\gamma$-reweight | 0.10% | 1.58% | 5.58% | 13.58% |
| STA($\gamma$=0.5) | 0.42% | 0.73% | 2.71% | 8.85% |
| SynthID | 0.52% | 2.60% | 4.79% | 9.90% |
| ITS-Edit | 0.00% | 0.52% | 1.67% | 3.75% |
| EXP-Edit | 0.31% | 8.96% | 15.10% | 19.17% |
| MCmark($n$=10) | 0.52% | 13.89% | 46.32% | 84.42% |
| MCmark($n$=20) | 1.15% | 18.44% | 43.02% | 70.94% |
| MCmark($n$=50) | 0.94% | 20.52% | 44.79% | 71.25% |
| MCmark($n$=100) | 0.73% | 17.29% | 34.06% | 56.77% |

Table 6: Robustness (TPR@5%FPR) of watermarking method across different attack types.

| Method | DIPPER | Random 30% | Random 20% | Random 10% |
|---|---|---|---|---|
| KGW($\delta$=0.5) | 1.67% | 14.89% | 19.79% | 21.60% |
| KGW($\delta$=1.0) | 2.08% | 30.73% | 47.81% | 61.67% |
| KGW($\delta$=1.5) | 4.38% | 58.13% | 77.71% | 89.17% |
| KGW($\delta$=2.0) | 5.63% | 80.74% | 91.16% | 96.00% |
| Unigram($\delta$=0.5) | 53.75% | 67.81% | 78.85% | 83.96% |
| Unigram($\delta$=1.0) | 60.83% | 84.17% | 92.19% | 91.46% |
| Unigram($\delta$=1.5) | 63.33% | 92.84% | 94.32% | 97.37% |
| Unigram($\delta$=2.0) | 74.79% | 98.23% | 99.48% | 100.00% |
| DIP($\alpha$=0.3) | 2.29% | 7.02% | 16.91% | 36.91% |
| DIP($\alpha$=0.4) | 1.77% | 7.79% | 18.21% | 39.16% |
| $\gamma$-reweight | 1.88% | 11.37% | 23.68% | 46.84% |
| STA($\gamma$=0.5) | 6.88% | 15.52% | 28.23% | 45.63% |
| SynthID | 9.90% | 23.13% | 32.60% | 49.90% |
| ITS-Edit | 5.00% | 8.23% | 10.73% | 11.88% |
| EXP-Edit | 5.10% | 25.73% | 29.69% | 39.38% |
| MCmark($n$=10) | 15.21% | 61.26% | 90.21% | 99.05% |
| MCmark($n$=20) | 13.85% | 58.65% | 79.90% | 94.38% |
| MCmark($n$=50) | 10.83% | 56.46% | 78.23% | 91.77% |
| MCmark($n$=100) | 10.63% | 50.63% | 68.85% | 83.85% |

Table 7: Unbiasedness, detectability, and robustness scores of watermarking methods, sorted by Detectability score.

| Method | Unbiasedness | Detectability | Robustness |
|---|---|---|---|
| SynthID | 0.965 | 0.974 | 0.105 |
| MCmark($n$=10) | 0.985 | 0.971 | 0.423 |
| MCmark($n$=20) | 0.988 | 0.959 | 0.390 |
| MCmark($n$=50) | 0.989 | 0.945 | 0.398 |
| KGW($\delta$=2) | 0.884 | 0.925 | 0.533 |
| MCmark($n$=100) | 0.985 | 0.906 | 0.330 |
| Unigram($\delta$=2) | 0.872 | 0.903 | 0.855 |
| Unigram($\delta$=1.5) | 0.927 | 0.838 | 0.711 |
| KGW($\delta$=1.5) | 0.935 | 0.808 | 0.350 |
| KGW($\delta$=1) | 0.972 | 0.724 | 0.155 |
| Unigram($\delta$=1) | 0.972 | 0.723 | 0.590 |
| DiPmark($\alpha$=0.5) | 0.980 | 0.719 | 0.078 |
| DiPmark($\alpha$=0.4) | 0.985 | 0.702 | 0.058 |
| EXP | 0.936 | 0.676 | 0.147 |
| STA($\gamma$=0.5) | 0.988 | 0.676 | 0.079 |
| DiPmark($\alpha$=0.3) | 0.991 | 0.660 | 0.051 |
| Unigram($\delta$=0.5) | 0.991 | 0.502 | 0.436 |
| ITS | 0.907 | 0.494 | 0.032 |
| KGW($\delta$=0.5) | 0.988 | 0.461 | 0.054 |

