# OpenReview forum: "Analyzing and Evaluating Unbiased Language Model Watermark"
_ICLR.cc/2026/Conference — ICLR 2026 Poster_

### Official Review · Reviewer_aAFa · 2025-10-30

**Soundness:** 2
**Presentation:** 3
**Contribution:** 2
**Rating:** 4
**Confidence:** 4

**Summary:**

This is a benchmark paper that evaluates different unbiased watermarking methods in large language models. Primally, the authors evaluate the detectability, unbiasedness, and robustness of different unbiased watermarking methods.

**Strengths:**

1. This paper introduces UWBench, a systematic evaluation of unbiased watermarking methods over the three-axis protocol: unbiasedness, detectability, and robustness.
2. The authors provide a better procedure to evaluate unbiasedness and robustness for unbiased watermarking methods, which I agree with.

**Weaknesses:**

1. The authors may miss some unbiased watermark baselines in the evaluation, e.g., [1], although this work is mentioned in the introduction section.
2. I believe the main issue with this paper is the lack of motivation to provide a benchmark work for unbiased watermarking methods. The authors should better justify why such a benchmark is necessary and what specific gaps it aims to fill in the existing literature. Without a clear motivation, the contribution appears limited.
3. A conceptual evaluation of different baselines could also be provided. For example, what are the drawbacks of each method? The authors could also offer guidelines for designing future unbiased watermarks.
4. The authors claim that UWBench is an open-source benchmark. I would appreciate it if the authors could provide the anonymized codes in their supplementary material.

[1] Christ, M., Gunn, S., & Zamir, O. (2024, June). Undetectable watermarks for language models. In The Thirty Seventh Annual Conference on Learning Theory (pp. 1125-1139). PMLR.

**Questions:**

Please see weaknesses.

---

> ### Author Response · Authors · 2025-11-21
> **Official Rebuttal by Authors [Part 1/2]**
>
> Thank you for your valuable feedback. We address your concerns below:
>
> > W1. The authors may miss some unbiased watermark baselines in the evaluation, e.g., [1], although this work is mentioned in the introduction section.
>
> A1. We thank the reviewer for raising this point. Christ et al. [1] is indeed an important theoretical contribution; however, it is a purely theoretical framework with **no available implementation**. This limitation has also been noted in subsequent unbiased watermarking works, including ITS-Edit (Kuditipudi et al.) and DiPmark (Wu et al.), neither of which provide empirical comparisons to Christ et al. for the same reason. Since no open-source or reproducible implementation exists, it is unfortunately not possible to include it as an empirical baseline in our evaluation. We will clarify this in the revised manuscript.
>
>
> > W2. I believe the main issue with this paper is the lack of motivation to provide a benchmark work for unbiased watermarking methods. The authors should better justify why such a benchmark is necessary and what specific gaps it aims to fill in the existing literature. Without a clear motivation, the contribution appears limited.
>
> A2. We kindly point the reviewer to Section 3.1, where we explicitly discuss the motivation for providing a dedicated benchmark for unbiased watermarking. Statistical watermarking is a task-agnostic verification method that applies to any language model and any downstream task without requiring new datasets. Therefore, the central challenge is not data collection but the absence of principled and reliable metrics for evaluating unbiased watermarking methods. Existing approaches assess watermarks along three axes—unbiasedness, detectability, and robustness—but only detectability currently benefits from well-established evaluation practices. Unbiasedness has so far been evaluated only under single-prompt settings, which miss important failure modes; in fact, we prove a fundamental impossibility result showing that no watermark can remain unbiased when the same prompt is queried repeatedly, motivating a new metric that measures distributional drift across generations. Robustness evaluation is also limited: paraphrasing-based attacks exhibit high variance and inconsistent behavior, as shown in Figure 2. To address this, our benchmark combines paraphrasing attacks with controlled token-modification attacks to provide stable and reproducible robustness measurements. In summary, UWBench fills a clear gap in the literature by shifting watermarking evaluation away from task-specific datasets and toward theoretically grounded, reproducible, and holistic metrics that more accurately characterize the strengths and limitations of unbiased watermarking algorithms.

---

> ### Author Response · Authors · 2025-11-21
> **Official Rebuttal by Authors [Part 2/2]**
>
> > W3. A conceptual evaluation of different baselines could also be provided. For example, what are the drawbacks of each method? The authors could also offer guidelines for designing future unbiased watermarks.
>
> A3. We provide a conceptual evaluation of the unbiased watermark baselines below.
>
>
> | Method            | Unbiasedness | Detectability | Robustness |
> |-------------------|--------------|---------------|------------|
> | MCmark (n=20)     | ⭐⭐⭐⭐☆       | ⭐⭐⭐⭐⭐        | ⭐⭐⭐⭐⭐     |
> | MCmark (n=10)     | ⭐⭐⭐⭐☆       | ⭐⭐⭐⭐⭐        | ⭐⭐⭐⭐☆     |
> | SynthID           | ⭐⭐☆☆☆       | ⭐⭐⭐⭐⭐        | ⭐⭐⭐⭐☆     |
> | γ-reweight        | ⭐⭐⭐☆☆       | ⭐⭐⭐☆☆        | ⭐⭐⭐☆☆     |
> | STA-1             | ⭐⭐⭐☆☆       | ⭐⭐☆☆☆        | ⭐⭐☆☆☆     |
> | DiPmark (α=0.4)   | ⭐⭐⭐⭐☆       | ⭐⭐☆☆☆        | ⭐⭐⭐☆☆     |
> | EXP-edit          | ⭐⭐⭐⭐☆       | ⭐⭐⭐☆☆        | ⭐⭐☆☆☆     |
> | ITS-edit          | ⭐⭐⭐⭐☆       | ⭐☆☆☆☆        | ⭐☆☆☆☆     |
> | Unigram (δ=2)     | ⭐☆☆☆☆       | ⭐⭐⭐⭐☆        | ⭐⭐⭐☆☆     |
> | KGW (δ=2)         | ⭐☆☆☆☆       | ⭐⭐⭐⭐☆        | ⭐⭐☆☆☆     |
>
>
> **DiPmark:** DiPmark maintains near-perfect unbiasedness but still sits at a **moderate detectability level**. This reflects the key limitation of unbiased, low-strength reweighting: the watermark remains too weak to detect reliably under adversarial edits.
>
>
> **γ-reweight and STA-1:** These heuristics improve detectability slightly over DiPmark but introduce more distributional drift, harming unbiasedness. They represent intermediate points in the trade-off space: stronger than DiPmark, but less principled and less robust.
>
>
> **ITS-edit:** While ITS-edit maintains reasonable unbiasedness, it suffers from both low detectability and low robustness. This baseline highlights that editing-based schemes often produce signals too weak or too brittle to survive perturbation.
>
>
> **EXP-edit:** EXP-edit reaches moderate detectability but shows weaker robustness, and its unbiasedness is noticeably below the top-performing methods. This indicates sensitivity to perturbations introduced during generation.
>
>
> **SynthID:** SynthID achieves high detectability and strong robustness, but at the cost of a noticeable drop in unbiasedness, supporting prior findings that SynthID-style logits boosting introduces systematic distributional bias.
>
>
> **MCmark:** MCmark has excellent detectability, strong robustness, and high unbiasedness, dominating along multiple axes. Its drawback, however, is computational cost and the need for multiple masked constraints, which may limit deployment efficiency.
>
> Our experiments shows two principles for designing effective unbiased watermarking methods:
>
> a) Increase signal strength without introducing large bias. Approaches like MCmark achieve high detectability by amplifying watermark signal in structured ways, suggesting that structured aggregation is more effective than naive boosting.
>
> b) Avoid strong hard-coded token preferences. Methods such as KGW and Unigram show that aggressive logit modifications harm unbiasedness. Future designs should keep per-token effects small and symmetric.
>
>
> > W4. The authors claim that UWBench is an open-source benchmark. I would appreciate it if the authors could provide the anonymized codes in their supplementary material.
>
> A4. We are currently cleaning and organizing the codebase and plan to fully open-source it by the end of the year. To meet the reviewer’s request during the review period, we have uploaded the complete source code as a ZIP file in the Supplementary Material.

---

> > ### Comment · Reviewer_aAFa · 2025-11-25
> >
> > I appreciate the authors' efforts in publishing their code and clarifying the work by Christ et al. I have raised my score to 6.

---

> > > ### Author Response · Authors · 2025-11-26
> > > **Author's response**
> > >
> > > Thank you again for the valueable comments and for taking the time to revisit our submission. We appreciate your recognition of our clarifications and the release of our code, and we are grateful that you raised your score.

---

### Official Review · Reviewer_xY8q · 2025-11-01

**Soundness:** 2
**Presentation:** 2
**Contribution:** 2
**Rating:** 2
**Confidence:** 4

**Summary:**

The paper proposed an evaluation framework for watermarking methods. The paper focuses on unbiasedness, detectability, and robustness. For unbiasedness, the paper proposed the Single-prompt multi-generation (SPMG) unbiasedness metric, which measures the difference of some metrics of generations from two models averaged across prompts and multiple generations of the same prompts. For robustness, the paper discussed the token effect region and the expected score decrease under one edit. In experiments, the weighted average of the true positive rate under different false positive rates and different perturbations is used. The proposed metrics are evaluated on several unbiased and a few biased watermarking methods.

**Strengths:**

Evaluation of watermarking methods is mostly ad-hoc, developing standard benchmarks is a timely topic. The discussion on the unbiasedness and robustness of watermarking methods makes sense.

**Weaknesses:**

1. The paper claims to introduce an open-source benchmark, but no source code is provided.

2. The contributions are not clear. In Section 5.1, there are a lot hyperparameters in the scores that seem to come from nowhere, e.g., the weights in the weighted average for the robustness score and detectability score. Why are these choices or these scores good? I would expect much more experiments to demonstrate that these scores are robust to the choice of hyperparameters, or give consistent evaluations for watermarks across different data sets, or number of prompts and length of generations.

3. The organization of the paper is confusing in some places, e.g., in section 4.2 token effect region and expected score decrease under one edit are discussed, but in the experiments, it seems that these concepts are not used, the score used for robustness are TPR under different FPR and perturbations. And the paragraphs "Green-count detectors" and "SynthID-style bit tests" seem to be examples for "expected score decrease under one edit". But they are all listed as if they were parallel concepts.

Some minor issues:

There are some typos, e.g., in line 465, I think it is Figure (1). In line 229, the form of $t_{\alpha}$ is inconsistent with the inequality above

Some citations are missing, e.g., for the DIPPER method in line 416, there are no citations.

**Questions:**

See weakness above

---

> ### Author Response · Authors · 2025-11-21
> **Official Rebuttal by Authors**
>
> Thank you for your valuable feedback. We address your concerns below:
>
> > W1. The paper claims to introduce an open-source benchmark, but no source code is provided.
>
> A1. We are currently cleaning and organizing the codebase and plan to fully open-source it by the end of the year. To meet the reviewer’s request during the review period, we have uploaded the complete source code as a ZIP file in the Supplementary Material.
>
> > W2. The contributions are not clear. In Section 5.1, there are a lot hyperparameters in the scores that seem to come from nowhere, e.g., the weights in the weighted average for the robustness score and detectability score. Why are these choices or these scores good? I would expect much more experiments to demonstrate that these scores are robust to the choice of hyperparameters, or give consistent evaluations for watermarks across different data sets, or number of prompts and length of generations.
>
> A2. Our primary contribution is the benchmark itself, namely, the three-axis evaluation protocol and the empirical results in Tables 1–4, aggregated across 7 datasets and 3 LMs. Section 5.1 serves only as a high-level summary that condenses these detailed results into a single visual illustration (Figure 1). The weights used in the aggregated scores were lightly tuned to (i) normalize each metric to the [0,1] range and (ii) roughly reflect the relative difficulty of the underlying tasks. Since this aggregated score is not used for evaluation conclusions, and all comparisons are shown directly using Tables 1–4, we believe extensive hyperparameter sensitivity analysis would not meaningfully strengthen the paper. We will clarify that the summary plot is purely illustrative and that the benchmark results are shown by the full metrics reported in Tables 1–4.
>
> > W3. The organization of the paper is confusing in some places, e.g., in section 4.2 token effect region and expected score decrease under one edit are discussed, but in the experiments, it seems that these concepts are not used, the score used for robustness are TPR under different FPR and perturbations. And the paragraphs "Green-count detectors" and "SynthID-style bit tests" seem to be examples for "expected score decrease under one edit". But they are all listed as if they were parallel concepts.
>
> A3. The purpose of Section 4.2 is to theoretically support our key claim: for unbiased watermarking, robustness fundamentally depends on the **number of tokens modified**, rather than on the specific attack type (paraphrasing, translation, synonym substitution, etc.). The notions of the token-effect region and the expected score decrease under a single edit are analytic tools used to reason about this phenomenon. In the experiments, we follow the standard robustness-evaluation protocol using TPR@FPR under various perturbation levels. We agree that the organization of Section 4.2 can be improved, and we will revise the section to make the relationship between theory and empirical evaluations clearer and to avoid confusing parallelism in the presentation.
>
> Minor issues: Thank you for pointing out these issues, we will fix them in the next revision.

---

> > ### Comment · Reviewer_xY8q · 2025-11-26
> > **Response to Rebuttal**
> >
> > I appreciate the authors' response. The source code is provided; for that, I am willing to increase my score to 4. I agree that in Table 1-4, for each evaluation criterion, a fair amount of experiments are provided. But still, there are methods that behave better on one criterion, but worse on the others. This would still be a pain point when comparing watermarking methods using the proposed benchmark. If some aggregated metrics, such as those discussed in Section 5.1, could give a consistent comparison for watermarking methods, that could be a great contribution, but the current aggregation across different criteria still seems to be a bit ad hoc.

---

> ### Author Response · Authors · 2025-11-26
> **Author's response**
>
> We sincerely thank the reviewer for the constructive feedback and for increasing the score. We also appreciate the acknowledgement of the comprehensiveness of our experimental evaluations in Tables 1–4.
>
> Regarding the reviewer’s concern that within each axis (detectability, unbiasedness, and robustness), different methods may perform better on one evaluation criterion but worse on others, we would like to offer the following clarification:
>
> Our benchmark is explicitly designed to capture the **multi-faceted nature** of each axis. For example, detectability includes both AUROC and TPR@FPR; unbiasedness includes both performance score and perplexity change; robustness includes multiple attack settings. These criteria are intentionally complementary rather than redundant. It is expected and in fact informative that a watermark may excel on one criterion within an axis but degrade on another, because each criterion captures a different aspect of the same axis.
>
> This is precisely why we did not focusing on developing advanced method to collapse each axis into a single “score”: doing so would implicitly impose subjective weightings and obscure the trade-offs that are central to understanding watermark behavior, such as a method performing well on single-generation unbiasedness but poorly on repeat-generation, or being robust to one type of attack but fragile to another.
>
> Reporting multiple complementary criteria provides diagnostic insight into why a method succeeds or fails, avoids misleading offsets in aggregation, and supports fair comparison across fundamentally different watermarking mechanisms. It also better reflects real-world needs, where different applications may prioritize different aspects of detectability, unbiasedness, or robustness. For these reasons, we treat aggregation as an optional prototype (Section 5.1), while multi-criterion evaluation remains the core of our benchmark.
>
> We also appreciate the reviewer’s point about the potential value of more principled score aggregation. It definitely provides additional value beyond the multi-criterion scores. In Section 5.1, we provide a first-step draft toward aggregated scoring by proposing simple and interpretable weights that reflect the relative importance of different criteria. Designing a consistent aggregated metric requires domain-informed weighting or multi-objective optimization remains an open research direction. In the revision, we will expand our discussion to (1) better emphasize why per-criterion reporting is necessary and (2) outline possible future directions for more structured aggregation across criteria within each axis.
>
> Thank you again for the constructive comments and for recognizing the importance of our benchmark.

---

### Official Review · Reviewer_QDvf · 2025-11-01

**Soundness:** 3
**Presentation:** 3
**Contribution:** 2
**Rating:** 6
**Confidence:** 3

**Summary:**

This paper introduces UWBENCH, a benchmark and theoretical framework for evaluating unbiased watermarking methods in large language models.
The authors propose a single-prompt multi-generation (SPMG) metric to quantify distributional drift under repeated queries and define a variance-controlled detection statistic using McDiarmid’s inequality to derive finite-sample guarantees.
On the empirical side, they benchmark several watermarking schemes across three axes — unbiasedness, detectability, and robustness — and argue that token-modification attacks offer more stable robustness evaluation than paraphrasing.
Overall, the work aims to provide a reproducible, theoretically grounded evaluation suite for future watermark research.

**Strengths:**

### Strengths

1. Clear motivation and structure. The paper identifies a real gap — the lack of principled evaluation for unbiased watermarking — and builds both theoretical and empirical tools to address it.
2. Balanced theoretical and empirical contributions. The impossibility result (Theorem 4.1) and the finite-sample McDiarmid bound (Theorem 4.2) are neat results that add formal grounding to an otherwise empirical area.
3. Reproducible benchmark. Making UWBENCH open-source with standardized metrics is a strong community contribution.
4. Thoughtful experimental framing. The move away from paraphrasing attacks toward token-modification attacks as robustness tests is well-justified and supported by variance analysis.
5. Readable figures and tables. The scatterplot (Fig. 1) effectively summarizes the trade-offs across unbiasedness, detectability, and robustness.

**Weaknesses:**

### Weakness

- While I found the theoretical framing around McDiarmid’s inequality and the single-prompt multi-generation (SPMG) detection metric interesting, the use of the concentration bound feels a bit “black-boxed.” The paper could be clearer about the assumptions that allow McDiarmid to hold — namely independence across samples and bounded differences — rather than invoking it as a matter of fact. Clarifying these assumptions would help readers trust the finite-sample guarantees and better understand what breaks when they are violated.

- Beyond that, I think the work would benefit from being placed more explicitly in the growing literature studying watermark–utility trade-offs. For example, Downstream Trade-offs of a Family of Text Watermarks (Ajith et al., EMNLP Findings 2024) shows that even unbiased watermarks can hurt downstream task accuracy by 10–20 %, and WaterJudge: Quality-Detection Trade-off when Watermarking Large Language Models (Molenda et al., NAACL Findings 2024) quantifies how stronger watermarks improve detectability but reduce generation quality. Most relevant, Watermarking Degrades Alignment in Language Models (Verma et al., ICLR 2025) demonstrates that watermarking shifts alignment behaviors such as truthfulness, safety, and helpfulness, and mitigates this via an external reward model. Verma et al. (2025) analyze how Gumbel watermarking affects model alignment. They note that Gumbel watermarking produces the same output deterministically for a fixed seed, meaning that in your repeated-query setup the SPMG drift would be trivially zero by design rather than informative about unbiasedness. It might help to briefly acknowledge this point when interpreting your results on Gumbel watermarking. Verma et al. also show that watermarking tends to decrease reward-model scores, which could, in principle, serve as another lens for detection. If your benchmark already defines a bounded per-generation metric Met(⋅), it might be interesting to mention that a reward-based version of this metric could empirically capture the same drift your McDiarmid analysis predicts. Even a short comment or future-work note on whether these two detection views—statistical vs. reward-based—might align would make the discussion more rounded without expanding scope.

**Questions:**

- **Notation check:** Equation (2) likely intends
  $$t_\alpha = A\sqrt{\tfrac{12\ln(1/\alpha)}{mn}},$$
  not $A^2\sqrt{\cdot}$.
- **Terminology:** Define “SPMG” and “Met” explicitly on first mention; readers outside the immediate subfield may not recognize these acronyms.
- **Writing:** The proofs in Appendix B are clear and worth highlighting — consider moving a short intuitive summary of Theorem 4.2’s logic into the main text.
- **Question:** Have the authors considered testing whether UWBENCH metrics correlate with alignment or reward-model score improvement as in Verma et al.? Even a small pilot experiment could reveal whether detectability improvements coincide with improved alignment or if they diverge.


This is a well-executed and timely paper that provides both theoretical insights and practical tools for unbiased watermark evaluation. My main feedback is to clarify the assumptions behind the theoretical bound and to connect the benchmark more explicitly to alignment and downstream-utility studies. Adding or even briefly discussing a reward-model-based alignment ablation would substantially strengthen the paper’s impact.

---

> ### Author Response · Authors · 2025-11-21
> **Official Rebuttal by Authors [Part 1/2]**
>
> Thank you for your positive and valuable feedback. We address your concerns below:
>
> > W1. While I found the theoretical framing around McDiarmid’s inequality and the single-prompt multi-generation (SPMG) detection metric interesting, the use of the concentration bound feels a bit “black-boxed.” The paper could be clearer about the assumptions that allow McDiarmid to hold — namely independence across samples and bounded differences — rather than invoking it as a matter of fact. Clarifying these assumptions would help readers trust the finite-sample guarantees and better understand what breaks when they are violated.
>
> A1. Thanks for pointing this out, we will modify the claims for better clarification.
>
> > W2. Beyond that, I think the work would benefit from being placed more explicitly in the growing literature studying watermark–utility trade-offs. For example, Downstream Trade-offs of a Family of Text Watermarks (Ajith et al., EMNLP Findings 2024) shows that even unbiased watermarks can hurt downstream task accuracy by 10–20 %, and WaterJudge: Quality-Detection Trade-off when Watermarking Large Language Models (Molenda et al., NAACL Findings 2024) quantifies how stronger watermarks improve detectability but reduce generation quality. Most relevant, Watermarking Degrades Alignment in Language Models (Verma et al., ICLR 2025) demonstrates that watermarking shifts alignment behaviors such as truthfulness, safety, and helpfulness, and mitigates this via an external reward model. Verma et al. (2025) analyze how Gumbel watermarking affects model alignment. They note that Gumbel watermarking produces the same output deterministically for a fixed seed, meaning that in your repeated-query setup the SPMG drift would be trivially zero by design rather than informative about unbiasedness. It might help to briefly acknowledge this point when interpreting your results on Gumbel watermarking. Verma et al. also show that watermarking tends to decrease reward-model scores, which could, in principle, serve as another lens for detection. If your benchmark already defines a bounded per-generation metric Met(⋅), it might be interesting to mention that a reward-based version of this metric could empirically capture the same drift your McDiarmid analysis predicts. Even a short comment or future-work note on whether these two detection views—statistical vs. reward-based—might align would make the discussion more rounded without expanding scope.
>
> A2. We greatly appreciate the reviewer’s detailed discussion of related work. We would first like to clarify that Downstream Trade-offs of a Family of Text Watermarks (Ajith et al., EMNLP Findings 2024) focuses on KGW, EWD, and SIR, **all of which are biased watermarking schemes**. As a result, the observed 10–20% drops in downstream accuracy are expected and consistent with the known behavior of biased watermarks. We will make this distinction explicit in the revision to avoid confusion.
>
> We also appreciate the reviewer’s suggestion about reward-model scores. Since our benchmark already defines a bounded per-generation metric Met(⋅), integrating a reward-based measure is indeed a promising direction for capturing alignment-related drift that parallels our McDiarmid-based statistical analysis. While this lies slightly beyond the scope of the current work, we will mention in the discussion section that reward-based detection could be an interesting future extension, potentially bridging statistical and alignment-oriented perspectives on watermark-induced drift.
>
>
> Notation check: We will polish the paper based on the reviewer’s suggestions.
>
> > Q1. Question: Have the authors considered testing whether UWBENCH metrics correlate with alignment or reward-model score improvement as in Verma et al.? Even a small pilot experiment could reveal whether detectability improvements coincide with improved alignment or if they diverge.
>
> A. Yes, we test the safety score in Verma et al. The results are shown below. Most unbiased watermark have great safety, while surprisingly, SynthID, one of the  unbiased watermark with strongest detectability have poor safety performance.
>
> Safe rate (100 samples from SAP2000 datasets)
>
>
> | Watermark         | Llama-3.2-3B-Instruct | Qwen2.5-3B-Instruct |
> |------------------|-----------------------|----------------------|
> | NoWatermark      | 0.97                  | 0.66                 |
> | KGW (δ=2.0)      | 0.97                  | 0.65                 |
> | Unigram (δ=2.0)  | 0.97                  | 0.76                 |
> | DIP (α=0.4)      | 0.96                  | 0.73                 |
> | γ-reweight       | 0.97                  | 0.76  |
> | STA (γ=0.5)      | 0.95                  | 0.68       |
> | SynthID          | 0.89                  | 0.62                 |
> | ITS-Edit         | 0.95                  | 0.65                 |
> | EXP-Edit         | 0.96                  | 0.64                 |
> | MCmark (n=10)    | 0.97                  | 0.71                 |

---

> ### Comment · Reviewer_QDvf · 2025-11-24
> **Response to Rebuttal**
>
> I appreciate the author's thoughtful response to my review comments and the additional numbers on safety. I have raised my recommendation to an accept
> .

---

> > ### Author Response · Authors · 2025-11-26
> > **Author's response**
> >
> > We are very glad to hear that our responses have addressed your concerns, and we sincerely appreciate you taking the time to reevaluate our work and increase your rating. Thank you for your thoughtful engagement and support.

---

### Official Review · Reviewer_CHL5 · 2025-11-01

**Soundness:** 2
**Presentation:** 3
**Contribution:** 2
**Rating:** 4
**Confidence:** 4

**Summary:**

This paper introduces UWBENCH, an open-source benchmark for unbiased language‑model watermarks. The authors contribute (i) a single‑prompt multi‑generation (SPMG) metric and a calibrated statistic DetWmk to quantify drift that emerges when the same prompt is queried many times, (ii) an impossibility theorem showing that any detectable unbiased watermark cannot preserve the LM distribution under repeated queries with a fixed key, and (iii) a token‑level robustness analysis with a simple certificate based on effect regions. Empirically, the benchmark evaluates methods along unbiasedness, detectability, robustness across several LMs and tasks, and argues that there is a trade-off between (i) optimizing detectability and unbiasedness, (ii) robustness.

**Strengths:**

1. Theorems 4.1 is clean and establishes a fundamental limitation of detectable watermarks.
2. The evaluation covers multiple models/datasets and reports matched‑FPR operating points.
3. The three‑axis protocol is well‑specified for unbiased watermarking.

**Weaknesses:**

1. The proposed unbiasedness metric uses performance metrics as surrogate and might not guarantee Eq.(1). Thus it is misleading to call it unbiasedness. The benchmark lacks reasoning tasks where the accuracy is an important performance indicator.
2. The robustness framework is neat but may not capture common edits such as typo, translation attack, synonym substitution. Also Random token replacement lacks practicality and distort the output distribution.
3. Novelty is limited compared with existing results: similar arguments to Theorems 4.1 is already established in Christ and Gunn (2024) and the three-axis evaluation is common in MarkMyWords.

**Questions:**

1. If the watermark key is re‑sampled per generation (or per prompt), does SPMG drift vanish in expectation?
2. In practice, is the same key used multiple times for the same prompt?
3. SPMG requires ∣Met(g)∣≤A; yet the text cites perplexity as an example, which is unbounded. Clarify how you bound or clip PPL (or switch to bounded NLL)?

---

> ### Author Response · Authors · 2025-11-21
> **Official Rebuttal by Authors [Part 1/2]**
>
> Thank you for your valuable feedback. We address your concerns below:
>
> > W1. The proposed unbiasedness metric uses performance metrics as surrogate and might not guarantee Eq.(1). Thus it is misleading to call it unbiasedness.
>
> A1. In practice, directly comparing the full output distributions of large language models is infeasible. As a result, prior unbiased watermarking works (e.g., Hu et al., Wu et al., Chen et al.) consistently rely on downstream performance metrics, such as perplexity, BLEU, and BERTScore, as practical surrogates for assessing distributional preservation. Perplexity in particular measures the log-likelihood of generated text under the unwatermarked model, and thus provides a meaningful proxy for distributional deviation relevant to Eq. (1). We will add a discussion clarifying this relationship to avoid confusion and better contextualize why performance-based evaluations are standard for unbiased watermarking.
>
> > W2. The benchmark lacks reasoning tasks where the accuracy is an important performance indicator.
>
> A2. We appreciate the reviewer’s suggestion. In evaluating unbiased watermarking, prior work typically considers three axes: unbiasedness, detectability, and robustness (Hu et al., Wu et al., Chen et al.). Our goal is to provide a general and unified framework covering all three aspects. Reasoning tasks can be viewed as a subcategory of the unbiasedness axis, since they measure downstream performance impacts. Due to time and resource constraints, we were not able to evaluate every downstream task, including reasoning and alignment. However, we included several widely used metrics, BLEU, BERTScore, and perplexity, as representative downstream evaluations following standard practice in the literature (Hu et al., Wu et al., Chen et al.). We agree that expanding to reasoning tasks would further strengthen the benchmark, and we will incorporate this in future extensions.
>
> > W3. The robustness framework is neat but may not capture common edits such as typo, translation attack, synonym substitution. Also Random token replacement lacks practicality and distorts the output distribution.
>
> A3. One of our goals is to clarify misconceptions about robustness evaluation in unbiased watermarking. As discussed in Section 4.2, the core mechanism behind all unbiased-watermark attacks is token modification. Therefore, when attacks modify the same number of tokens, their impact on unbiased watermarks is fundamentally similar, regardless of whether the modification arises from paraphrasing, translation, synonym substitution, or typographical edits. Stronger attacks do not reflect methodological sophistication; they simply correspond to modifying more tokens and thus causing greater distributional drift. From the watermark designer’s perspective, robustness to a fixed amount of token-level perturbation generalizes across different editing styles. From the attacker’s perspective, attacks must simultaneously preserve quality, which severely restricts the number of tokens they can alter. We will clarify this motivation in the revised manuscript.
>
> > W4. Novelty is limited compared with existing results: similar arguments to Theorems 4.1 is already established in Christ and Gunn (2024) and the three-axis evaluation is common in MarkMyWords.
>
> A4. We would appreciate further clarification regarding the claimed similarity of Theorems 4.1 and Christ and Gunn (2024). Our understanding is that Christ and Gunn (2024) operate under a fundamentally different setting: computationally indistinguishable watermarks, which explicitly allow small biases in the output distribution. In contrast, our work studies strict unbiasedness, which requires exact preservation of the distribution in expectation. Theorem 4.1 is not intended as a novel contribution; rather, it supports our empirical observation that repeated generation under the same prompt can introduce detectable drift, even for nominally unbiased schemes.
>
> Regarding the three-axis evaluation, we agree that it is widely used in LM watermarking (e.g., MarkMyWords, DiPmark, MCMark). Our contribution lies in being the first to propose a dedicated benchmark tailored specifically for unbiased watermarking, empirically show the intrinsic bias in unbiased watermark with SPMG, and clarify discussions on unbiasedness and robustness in this setting.

---

> > ### Author Response · Authors · 2025-11-21
> > **Official Rebuttal by Authors [Part 2/2]**
> >
> > > Q1. If the watermark key is re‑sampled per generation (or per prompt), does SPMG drift vanish in expectation?
> >
> > A. If you are referring to re-sampling the secret key $sk$ for each generation (or each prompt), then yes, under this setting, the SPMG drift vanishes in expectation. However, using a fresh key per generation creates a different and more serious issue: it makes watermark detection practically infeasible. During detection, the verifier only sees the generated text and must infer whether any valid watermark key could have produced it. If each generation uses a unique, never-reused key, the detector has no way to know which key to test, and detection becomes impossible. Thus, unbiased watermarking methods intentionally reuse the same key across generations to maintain detectability.
> >
> > > Q2. In practice, is the same key used multiple times for the same prompt?
> >
> > A. Yes. In unbiased watermarking practice, the same secret key $sk$ is reused across generations, even for repeated prompts. This reuse is crucial for efficient and reliable detection, as it allows the detector to apply a single known key rather than searching over an unbounded key space.
> >
> > > Q3. SPMG requires ∣Met(g)∣≤A; yet the text cites perplexity as an example, which is unbounded. Clarify how you bound or clip PPL (or switch to bounded NLL)?
> >
> > A. Thanks for pointing out this. In practice we cap the PPL under 50.

---

> > > ### Author Response · Authors · 2025-11-27
> > > **Additional Unbiasedness Experiments on Reasoning Tasks**
> > >
> > > Based on the reviewer’s suggestion, we provide additional reasoning-task evaluations using accuracy-based metrics. Specifically, we use the *NarrativeQA* dataset from HuggingFace. Following our unbiasedness evaluation protocol, we report accuracy under two settings:
> > >
> > > * 1000 prompts with one generation each, and
> > >
> > > * 10 prompts with 100 generations each.
> > >
> > > The results are presented in the tables below.
> > >
> > > | Watermark Processor (1000 prompts, 1 generation each)| Accuracy |
> > > |------------------------------------------------------|--------------------|
> > > | None                                                 | 0.7255             |
> > > | DiPmark(α = 0.4)                               | 0.7098             |
> > > | γ-reweight                                           | 0.7155             |
> > > | MCMark (n = 10)                                      | 0.7039             |
> > > | STA_1                            | 0.7143             |
> > > | SynthID(m = 30)                       | 0.7120             |
> > > | Unigram(δ = 2.0)                  | 0.6581             |
> > > | KGW(δ = 2.0)           | 0.6638             |
> > > | EXP_edit| 0.7066             |
> > > | ITS_edit   | 0.7167             |
> > >
> > > | Watermark Processor  (10 prompts, 100 generations each) | SPMG Accuracy |
> > > |------------------------------------------------------|-----------------------|
> > > | None                  | 0.048007              |
> > > | DiPmark(α = 0.4)                       | 0.041542              |
> > > | γ-reweight                                           | 0.042613              |
> > > | MCMark(n = 10)                                      | 0.030942              |
> > > | STA-1                            | 0.063040              |
> > > | SynthID(m = 30)                       | 0.028854              |
> > > | Unigram(δ = 2.0)                  | 0.090969              |
> > > | KGW(δ = 2.0)           | 0.080948              |
> > > | EXP_edit                          | 0.409834              |
> > > | ITS_edit                            | 0.394154              |
> > >
> > > We will also integrate the reasoning-task evaluations into our benchmark to ensure broader and more representative assessment of unbiasedness.

---

### Meta-Review · Area_Chair_qtji · 2026-01-06

**Summary:**

Reviewers consider this work a timely benchmark for unbiased watermarks. Strengths include the three-axis protocol, breadth of experiments, and framing around single‑prompt multi‑generation drift and token-level attacks. Initial concerns were raised about the validity of the “unbiasedness” proxy metrics, robustness realism, and theoretical clarity. The rebuttal addressed many of these concerns including (1) clarifying assumptions (2) provision of supplementary code and (3) additional evaluations including reasoning-task accuracy and an alignment/safety evals, and this was reflected in multiple reviewers increasing scores prior to the rollback period.

**Reviewer Concerns:**

The rebuttal addresses most of the substantive concerns: code, confusion about metrics, key reuse and interpretation of single-prompt multi-generation, clearer framing of Theorem 4.1 vs prior work, downstream/behavioral evals. One remaining concern is the desire for a single aggregate score, versus the multi-axis protocol proposed by the authors. I tend to agree with the authors perspective that multi-axis scoring is the right way to assess these watermarks.

**Reviewer Scores:**

CHL5 - did not engage; major concerns seem satisfactorily addressed in my opinion.

QDvf - already increased to 8 before the rollback.

xY8q - already raised to 4 before the rollback; likely remains borderline.

aAFa already raised to 6 before the rollaback.

---

### Decision · Program_Chairs · 2026-01-26

Accept (Poster)